# Optical Emission Spectroscopy of Underwater Spark Generated by Pulse High-Voltage Discharge with Gas Bubble Assistant

**Vitaliy Stelmashuk \*** , **Vaclav Prukner, Karel Kolacek** , **Andrii Tuholukov, Petr Hoffer, Jaroslav Straus, Oleksandr Frolov** and **Vit Jirasek**

Department of Pulse Plasma Systems, Institute of Plasma Physics of the Czech Academy of Sciences, Za Slovankou 1782/3, 182 00 Prague, Czech Republic; prukner@ipp.cas.cz (V.P.); kolacek@ipp.cas.cz (K.K.); tuholukov@ipp.cas.cz (A.T.); hoffer@ipp.cas.cz (P.H.); straus@ipp.cas.cz (J.S.); frolov@ipp.cas.cz (O.F.); jirasek@ipp.cas.cz (V.J.)
\* Correspondence: stelmashuk@ipp.cas.cz

**Abstract:** This paper is aimed at the investigation of the acoustic and spectral characteristics of underwater electric sparks generated between two plate electrodes, using synchronized gas bubble injection. There are two purposes served by discharge initiation in the bubble. Firstly, it creates a favorable condition for electrical breakdown. Secondly, the gas bubble provides an opportunity for the direct spectroscopy of plasma light emission, avoiding water absorption. The effect of water absorption on captured spectra was studied. It was observed that the emission intensity of the $H_a$ line and a shockwave amplitude generated by discharge strongly depend on the size of the gas bubble in the moment of the discharge initiation. It was found that the plasma in the underwater spark channel does not correspond to a source of black-body radiation. This study can be also very useful for understanding the difference between discharges produced directly in a liquid and discharges produced in gas/vapor bubbles surrounded by a liquid.

**Keywords:** water; underwater spark; optical emission spectroscopy; shockwave; black body radiation



## 1. Introduction

There is an increasing interest in understanding the nature of underwater discharges generated by high-voltage electrical pulses, and also in the study of associated phenomena: spark expansion in water, shockwave generation, dynamic of the cavitation bubble generated by the spark, UV radiation, and chemically active species (see, for example, References [1–8]). The underwater discharge develops in two compulsory phases, namely the pre-breakdown phase and the spark phase.

The pre-breakdown phase starts by the initiation of the streamers and their subsequent propagation. The initiation may occur in microbubbles permanently presented in the water, or artificially produced vapor bubble/cavity through Joule heating [9], by a shockwave [10–13] or by a gas injection [14]. Discharges in water are of interest because they generate chemically active species (e.g., OH and $H_2O_2$). This makes plasma in water particularly suitable for disinfection and decontamination [5]. Electric discharges in bubbles within liquids are used as a source of UV radiation, ozone, and hydrogen peroxide [15].

As soon as a streamer bridges the electrode gap, the electrical breakdown occurs, and an underwater spark is generated [11]. The expanding spark channel is filled with cold high-pressure weakly ionized plasma originating from the gas of the injected bubble and evaporated water. There are numerous publications devoted to the theoretical or experimental studies of hydrodynamic phenomena induced by underwater spark discharges (see, for example, References [8,16–21]). However, the study of plasma in the spark channel has so far received less attention. Experimental studies to date have mainly focused on optical emission spectroscopy (OES) of the streamer or corona discharges (see, for example, References [5,22]), and only two publications can be found dealing with the OES of underwater

spark discharge [23,24]. There are two major challenges that the OES of the spark is facing: (1) the strongly varying spectral light absorption of water in the visible range and (2) the explosive expansion of the spark channel. In this study, a new experimental setup utilizing an injected gas bubble was employed. This allowed us to capture the light emission spectra directly from the underwater spark, while eliminating the water absorption. It has been found that the spark characteristics strongly depend on the volume of the gas bubble in which the electric discharge is initiated. The method of this study can help us understand the difference between discharges produced directly in a liquid and discharges produced in gas/vapor bubbles surrounded by a liquid.

## 2. Experimental

The experimental apparatus is shown in Figure 1. A rod-to-rod electrode system is mounted in a tank filled with tap water (conductivity 400 µS/cm). The electrodes are made from brass and are 6 mm in diameter. The typical interelectrode gap is 7 mm. A through hole is embodied in the cathode with a diameter of 1.6 mm. The optical fiber is placed in the hole. The end of the fiber is located 4 mm from the hole outlet (Figure 1). In addition, the cathode has a gas inlet through the same hole used for nitrogen or argon injection. The gas is supplied from the cylinder to the inlet via an electromechanical valve and is used for a bubble injection from the cathode outlet into the inter-electrode space. The valve opening time is 100 ms. The gas overpressure before valve opening is 2 bar. The capacitor $C_s$ (0.8 µF) is charged to the voltage of 15 kV. A triggered spark gap switches the capacitor to the electrode system with adjustable delay with respect to the moment of gas bubble injection. The typical time delay between the gas injection and the spark gap triggering varies from 20 to 27 ms. The delay is controlled by the BNC 575 pulse generator. Temporal voltage and current waveforms are measured by the voltage divider PVM-4 2000:1 (North Star Research Co.) and the Pearson probe (model 101), respectively. The measured voltage and current waveforms are recorded by an oscilloscope (Tektronix MDO 4054C). The current channel serves as the trigger source of the oscilloscope. In this way the time base of this oscilloscope is synchronized with the breakdown moment, i.e., when the underwater spark is formed. The oscilloscope generates the output trigger signal that switches the high-speed camera and the spectrometer (Figure 1). The high-speed camera (Phantom v710 equipped with a Nikon 200 mm f/4D IF-ED AF Micro objective) is utilized to capture video sequences of the gas bubble injection and the dynamic of the electrical discharge evolution. The processes are visualized by using the optical shadowgraphy method. A high-intensity light source operating in a continuous regime (COOLH Dedocool tungsten light head) is used for this purpose. A multichannel detector system is used to register radiation emitted by the spark channel. The output of the fiber bundle is coupled to the iHR-320 (Jobin-Yvon, Edison, NJ, USA) imaging spectrometer (grating 300 G mm$^{-1}$, blazed) that decomposes the incoming light into the spectrum, which is registered (synchronously with the breakdown moment) by a fast DH740i-18U-03 iStar ICCD camera (Andor, Belfast, UK). The exposition time of this camera is 3 µs. All spectra are corrected for the apparatus function by using a calibration performed with a broadband emission lamp (DH-2000-CAL). The absorption UV–VIS spectrum of deionized water was measured by a double-beam UV/VIS absorption spectrophotometer Specord 210 Plus (Analytik Jena, Jena, Germany) in a quartz cuvette with a 1 cm pathlength. The absorbance was calculated by the instrument against the reference spectrum measured with the empty cuvette. Finally, a correction for reflection on two interfaces of quartz–water was applied. The spark discharge generates a shockwave in surrounding water, the pressure of which is detected by a Muller-Platte needle probe, with the detection frequencies ranging from 0.3 to 11 MHz. The tip of the probe is positioned at a distance of 70 mm from the electrode gap. The measured acoustical signal is recorded by an oscilloscope (Tektronix TDS3054C). In order to minimize an electromagnetic interference from the discharge, this oscilloscope is powered from a backup power supply (EATON 5E 850i).

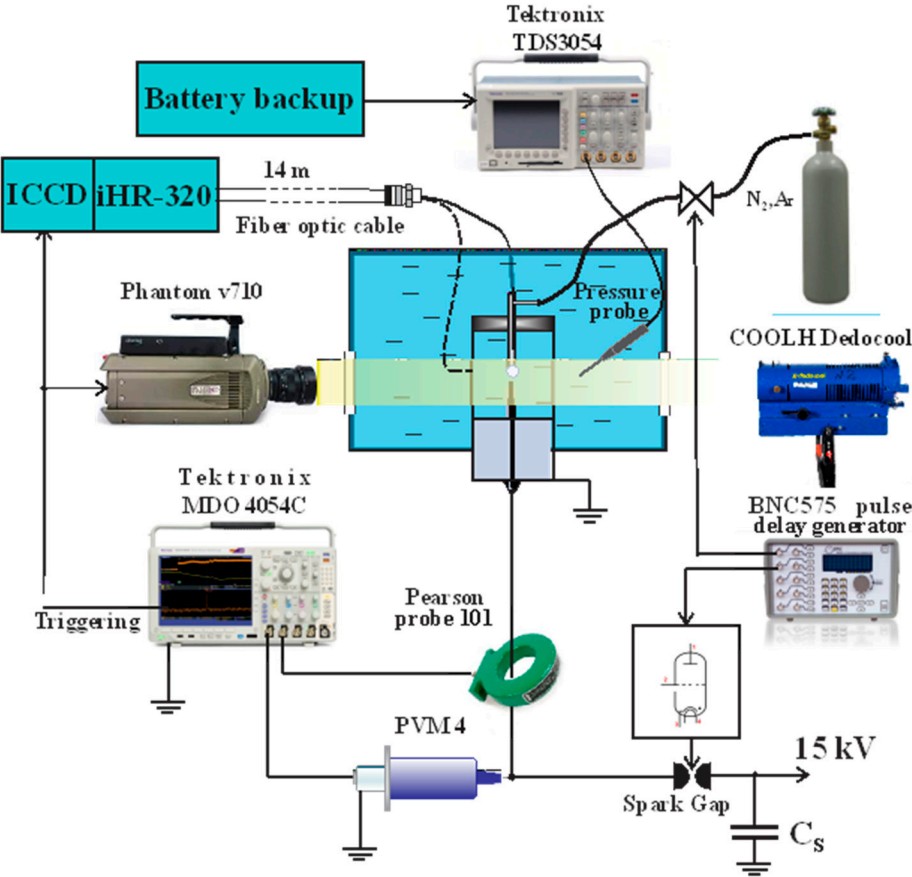

**Figure 1.** Experimental setup for spectroscopy investigation of spark discharge in water.

## 3. Results

A general description of the underwater discharge generated between plate-to-plate electrodes with nitrogen bubble injection was given in Reference [14]. After voltage application, the streamer develops in the injected gas bubble as a result of an avalanche process and propagates along the surface of the bubble [14]. After the streamer reaches the anode side of the bubble, it continues to propagate in water. When it bridges the interelectrode gap, the electrical breakdown takes place, and the underwater spark is generated. During this process, the internal pressure in the bubble is higher than the hydrostatic pressure in the surrounding water. This prevents the water penetration into the cathode through-hole and eliminates the effect of water absorption on the optical emission spectroscopy.

The spark optical emission is routed out of the cathode through-hole by using a quartz optical fiber. The aim is to demonstrate the effect of the presence of water on the measured optical spectra. Figure 2a provides a comparison of the measured optical emission spectra between 300 and 830 nm of two underwater electrical discharges generated with the help of a nitrogen gas bubble of the same size and of similar temporal voltage and current waveforms. The blue curve represents the emission spectrum obtained by the optical fiber placed in the water (dashed line in Figure 1), and the red curve represents the emission spectrum obtained by using optical fiber mounted in the cathode (Figure 1). The spectra were normalized for clarity. Both spectra are dominated by a hill-like region lying between 350 and 600 nm. The major difference between the spectra lies in the range from 480 to 720 nm, where the emission spectrum obtained by the optical fiber placed in water is partially absorbed. Particularly, the peak around 656 nm, which corresponds to the excited state of atomic hydrogen, $H_a$, is weak in comparison with the emission spectrum collected directly from the spark channel. It should also be noted that the difference between the measured spectra is expectable according to the measured absorption spectrum of tap water

used in the experiment (Figure 2b). However, the original idea to explain this difference by water absorption failed. It turned out to be negligible at a layer of water with thickness of units of centimeters (Figure 2b). There are several possible explanations for absorption enhancement: an absorption by a shockwave generated by a spark expansion, a reflection on a spark channel and/or its surface irregularities [10–13], or an absorption by vapor layer [10].

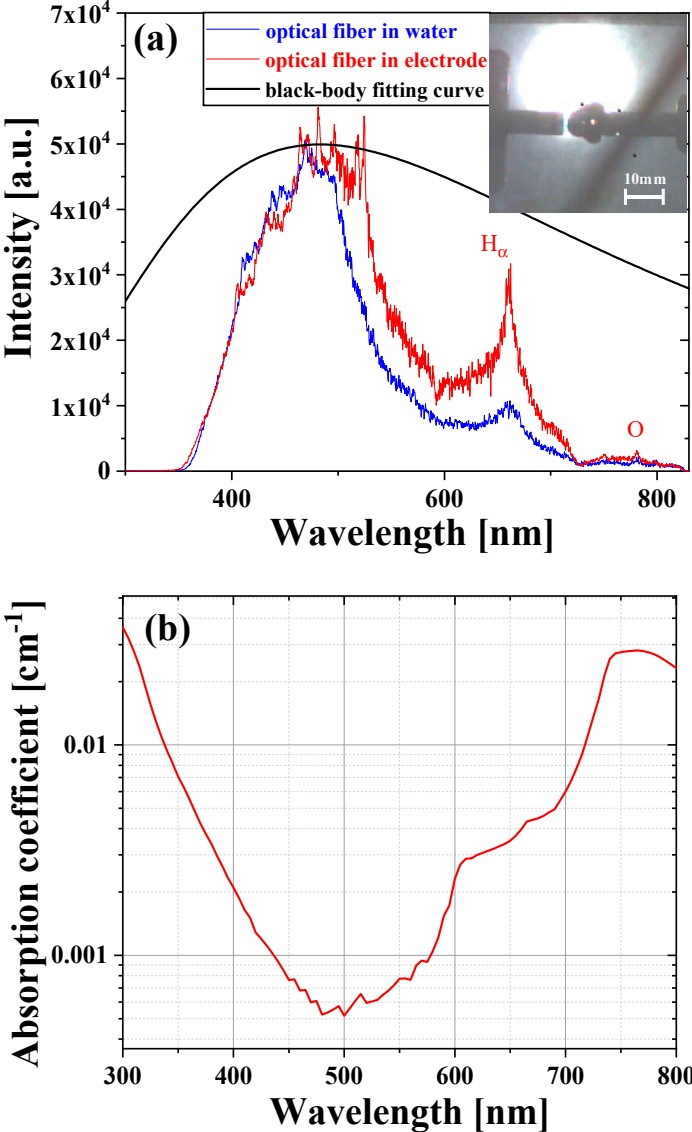

**Figure 2.** (**a**) Optical emission spectra of underwater spark demonstrating the effect of water on spectroscopy measurements and black-body radiation fitting curve. Inset shows the image of injected nitrogen bubble before the breakdown. (**b**) Measured optical absorption of used tap water.

There are some publications considering plasma in underwater sparks as a source of black-body radiation inherently (see, for example, References [18–20]). Unfortunately, the unambiguous fitting of experimental spectra to the black-body radiation curve is still missing. To check whether the light emitted by the underwater spark originates from a black-body source, the Planck's radiation law is fitted to the spectrum shown in Figure 2, assuming the temperature in the spark channel to be 6000 K. The fitting procedure shows that the black-body curve does not fit the observed results. Therefore, we must conclude that the underwater spark is not a source of a black-body radiation. Nevertheless, the assumption of black-body emission by the spark is able to simplify the mathematical model

in some cases and can give a reasonable result in hydrodynamic mathematical simulations of underwater spark expansion [16–21].

The electrical and optical emission properties of reported underwater discharges depend on the size of the injected gas bubble in the moment of the spark initiation. Figure 3 shows examples of current and voltage waveforms of three typical discharges differing by the size of injected nitrogen bubble just before the spark initiation: (*a*) the injected bubble almost interconnects the electrodes, (*b*) the bubble size is around of 50% of interelectrode gap, and (*c*) the bubble size is approximately 30% of interelectrode gap. The time interval between the voltage application and electrical breakdown depends on the distance between the front of the bubble and the counter electrode. Therefore, with the shortening time delay between the bubble injection and the voltage application, the spark channel in liquid is getting longer, and the spark channel in the bubble is becoming shorter; simultaneously, the delay between the voltage application and the breakdown is becoming longer. Unfortunately, the video frames corresponding to the time period of spectroscopy measurements are overexposed and cannot be presented. Instead, the images of the bubble just before the breakdown are presented in the insets of Figure 3. Figure 3d in shows the spark in 48 μs after the breakdown of the discharge identical to that with the voltage current characteristic presented in Figure 3a.

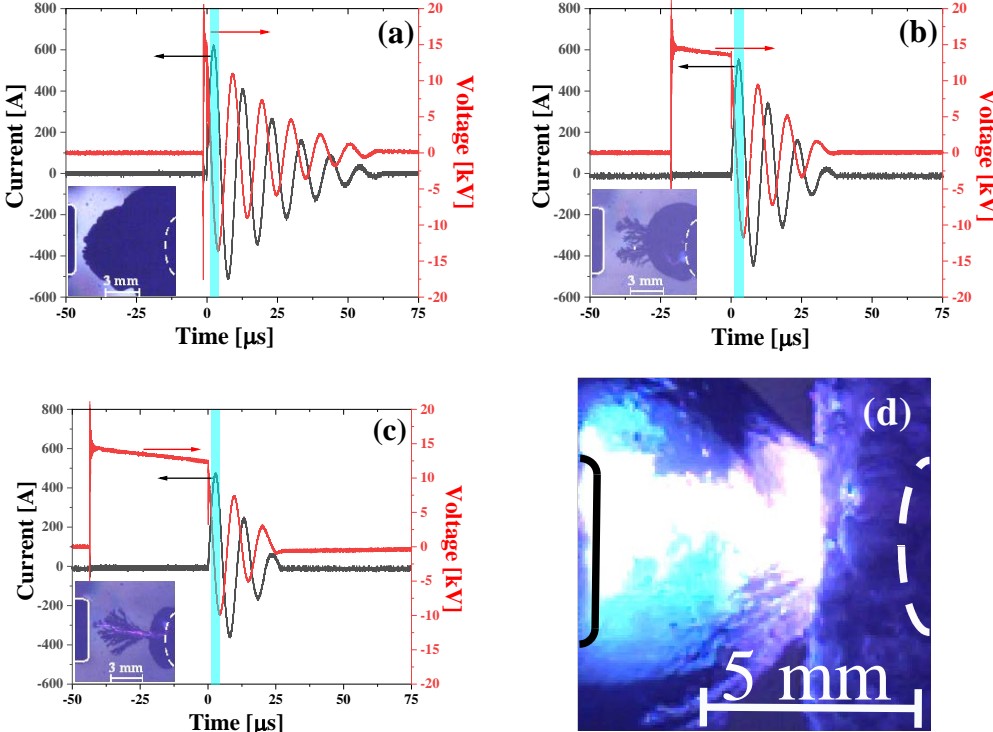

**Figure 3.** (**a–c**) Temporal voltage and current waveforms of underwater discharges initiated in nitrogen bubbles of different sizes. Insets in images show the images of the injected nitrogen bubble just before the breakdown (exposure time 0.4 μs); the surface of the anode is outlined by a white solid line, while the white dashed line depicts the cathode surface; the cyan area represents the exposure time period during which the optical emission spectra were measured. (**d**) Spark generated in nitrogen bubble identical to the discharge shown in image (**a**) captured 48 μs after the gap breakdown (exposure time 0.4 μs); black solid line contour shows the anode surface, and white dashed line shows the cathode surface.

Figure 4 shows the emission spectra of light emitted by sparks (with the corresponding voltage and current waveforms shown in Figure 3) transmitted by the optical fiber mounted in the hollow electrode (Figure 1). The origin of the strong light emission in the range from 350 to 580 nm is unclear. One can assume that the emission in this spectral region

is the pressure-broadened line radiation of hydrogen (H$_\beta$ {486.1 nm}, H$_\gamma$ {434.0 nm}, and H$_\delta$ {410.2 nm}), and oxygen ((2p)$^1$S-(2p)$^1$D{557.7 nm}) from de-composed water and band/line radiation of molecular/atomic nitrogen from bubble gas. Besides that, an oxygen line (3p)$^5$P–(3s)$^5$S {777.4 nm} appears even on the long-wavelength wing of the registered spectrum.

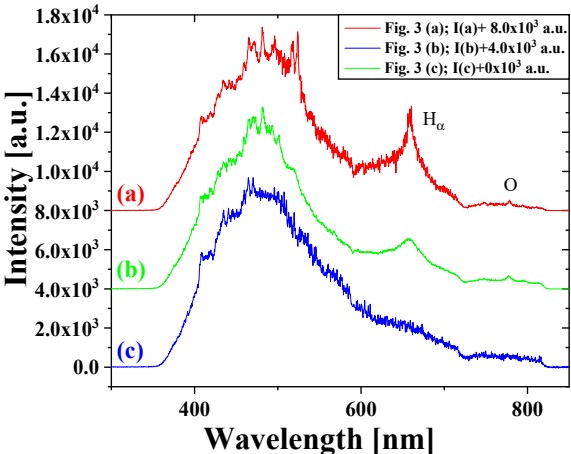

**Figure 4.** Optical emission spectra obtained from the discharge initiated in the different sizes of nitrogen bubble, using the optical fiber mounted in the cathode. The individual spectra are normalized (multiplied to have the same maximum value) and shifted vertically.

The shape of the H$_\alpha$ line {656.3 nm} directly correlates with the size of injected nitrogen bubble at the electrical breakdown moment (see Figure 4)—being more apparent at larger bubble sizes. Our interpretation is very simple: the larger the injected bubble is in the breakdown moment, the shorter the path of the discharge in water is, and the smaller the amount of water that is evaporated/de-composed, the smaller the pressure in the discharge channel is and, therefore, the narrower and more well pronounced the H$_\alpha$ is. On the contrary, the smaller the injected bubble is in the breakdown moment, the longer is the path of the discharge in the water, and the larger the amount of water that is evaporated/de-composed, the higher the pressure in the discharge channel is and, therefore, the broader and less pronounced (more buried in the background) the H$_\alpha$ is. The idea of the pressure broadening of the H$_\alpha$ line is supported by an independent pressure measurement of the pressure wave at a radial distance of 70 mm from the discharge by the Muller-Platte needle probe, performed simultaneously with spectroscopic measurements (see Figure 5). A similar effect was observed by Gamaleev et al. [24], who investigated an underwater micro-arc discharge in a pressurized chamber at different pressures. They reported an increase of continuum radiation of micro discharge plasma in seawater and disappearance of the H$_\alpha$ line with the increasing pressure. In their experiments, the H$_\alpha$ line is dominant in the spectrum obtained at atmospheric pressure (0.1 MPa). At pressures of 15 and 17.5 MPa, the continuum emission intensity increases in a wide range of wavelength, and the H$_\alpha$ peak almost disappears [24]. The spectra measured at high pressures by Gamaleev et al. closely resemble the spectrum obtained by optical fiber at small bubble/high pressure (Figures 2 and 4c). Therefore, it unambiguously indicates that the high pressure is a real cause of continuum emission in the range from 350 to 580 nm, regardless of whether or not the pressure is the result of spark heating by electrical currents and hydrodynamic processes (as in our case) or the discharge is generated in the pressurized environment (the case of Gamaleev et al. [24]).

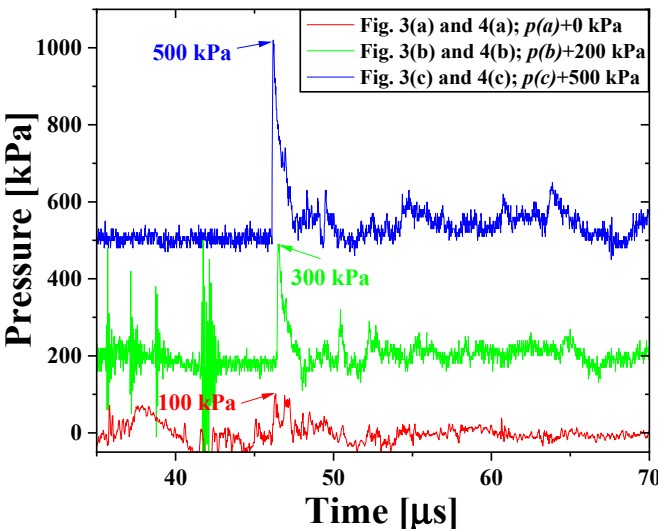

**Figure 5.** Pressure waveforms of a shockwave as a function of time.

The experiments described above were carried out by using argon as injected gas, as well. The obtained spectra appear to be identical to those obtained by using nitrogen.

## 4. Conclusions

Plasma in underwater spark discharge generated between two rod electrodes was investigated by means of OES. The negative effect of light absorption on the collecting optical spectra was demonstrated. The effective method of OES measurements, allowing us to avoid the negative effect of light absorption, was proposed, and the spectra of spark plasma not affected by the absorption effect were obtained. The result of the OES demonstrates that the plasma in underwater spark does not have a characteristic of black body source. The dominant hill-like region in the range from 350 to 580 nm may be a result of substantially broadened line radiation. The $H_\alpha$ line tends to broaden (from FWHM {full width at half magnitude} 21.6 nm {discharge a)}, through 35.2 nm {discharge b)} to 60.8 nm {discharge c)} with the gas bubble diminution (from bubble taking 90% of the gap {discharge a)}, through 50% {discharge b)} to 30% {discharge c)} at discharge initiation instant (due to pressure increase in the spark channel).

**Author Contributions:** Conceptualization, V.S.; methodology, V.S., P.H., V.P. and J.S.; validation, V.S., V.J. and K.K.; formal analysis, V.S. and K.K.; investigation, V.S. and A.T.; resources, V.S. and O.F.; data curation, V.P.; writing—original draft preparation, V.S.; writing—review and editing, V.S. and K.K.; visualization, V.S.; supervision, V.S.; project administration, V.S.; funding acquisition, V.S. and O.F. All authors have read and agreed to the published version of the manuscript.

**Funding:** Grant Agency of the Czech Republic (Grant No. GA18-12386S) and Grant Agency of the Ministry of Education, Youth and Sports of the Czech Republic (INTER-EXCELLENCE/Inter-Cost LTC20061).

**Institutional Review Board Statement:** Not applicable.

**Informed Consent Statement:** Not applicable.

**Data Availability Statement:** The data used in this paper are available from the author upon reasonable request.

**Conflicts of Interest:** The authors declare no conflict of interest.

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
