# Peer review of "Optical Emission Spectroscopy of Underwater Spark Generated by Pulse High-Voltage Discharge with Gas Bubble Assistant"

_processes, doi:10.3390/pr10081474_

Round 1

Reviewer 1 Report

This is a solid, well written paper, and it demonstrates good use of optical emission spectroscopy for investigation of underwater discharges. The methodology and the results are presented clearly. It seems rather obvious that the emission from the spark should not be a Planck function. But this makes the problem more interesting: more detailed analysis of the spectral lines (widths, relative intensities) may provide more information on the properties of the spark-produced plasma. Additionally, there is an interesting observation of pressure broadening of H-alpha line, it can be a subject of further investigations that are of interest for other applications. 

I can recommend this paper for publication in its present form.

Reviewer 2 Report

1. line 62, the gas speed and pulse high power frequency shall give

2. Fig. 2, what “hill”-like peak between 350 and 600 117 nm shall be discussed.

Line 126, what means the “absorption enhancement” in here, atomic hydrogen Ha? It shall be explained

What thickness of water used in Fig.2(b)?

3.Fig. 3 (a) the cathode was semi-circular but in (d) it was plate. It shall be identical.

The discharge voltages in three cases were different, did the energy of the capacitor was also changed with each other?

Line 160, the discharge period was smaller than 50μs, then how to get the image of Fig. 3(d) after 48s breakdown?

4. Fig.4, the base lines for three curves were quietly different, the data in y-axial shall be normalized. So was Fig. 5

The Hα line tends to broaden or heighten with the gas bubble diminution? It is better to give the full width of maximum height(FWMH) 

Fig. 4 the spectrum was obtained in the water or cathode electrode shall be denoted.

5. line 224, the Hα line tends to broaden with the gas bubble diminution at discharge initiation instant (due to pressure increase). Which was the high pressure? it might be mistake in the text.

Hα line was decreased, even disappeared with the increasing pressure, but in Fig. 4 it appeared in 100kPa, but disappeared in 300kPa.

6.why there was no the strongly spectral lines between 300-500nm even the OES was obtained in the cathode? As said the OES in the cathode was similar to the discharge in the air. In water and in the electrode, the OES was similar except the intensity.

7. line 220, conclusions, how to result that no light absorption effect in this work, you should give the OES in the air to compare them , and then you can give a conclusion.

The negative effect shall be defined in previous.
